# Outcome of Acquired Fanconi Syndrome Associated with Ingestion of Jerky Treats in 30 Dogs

**DOI:** 10.3390/ani12223192

**Published:** 2022-11-18

**Authors:** Stinna Nybroe, Charlotte R. Bjørnvad, Camilla F. H. Hansen, Tenna S. L. Andersen, Ida N. Kieler

**Affiliations:** 1Department of Veterinary Clinical Sciences, Faculty of Health and Medical Sciences, University of Copenhagen, 1870 Copenhagen, Denmark; 2Skibhus Dyreklinik, 5000 Odense C, Denmark; 3Dyrlæge Mikkelsen, 7400 Herning, Denmark

**Keywords:** canine, Fanconi, renal tubulopathy, proximal tubuli, glycosuria, jerky

## Abstract

**Simple Summary:**

Jerky induced Fanconi syndrome is a kidney disease related to ingestion of jerky treats, mainly in dogs, resulting in excretion of sugar in the urine despite normal blood sugar. The objectives of this study were to describe disease characteristics and long-term outcome. In 30 dogs with spontaneously occurring jerky induced Fanconi syndrome, common clinical signs were increased urination and drinking, tiredness and weight loss. Less common were decreased appetite, vomiting and diarrhea. Urine analysis often revealed blood and protein in the urine and some dogs had electrolyte disturbances evident from blood samples. Full clinical recovery was achieved for 79% of the dogs in 0.3–52 weeks (median 11 weeks) while sugar in the urine was resolved in 93% of the dogs within 1–31 weeks (median 6.5 weeks). These results indicate a more favorable outcome of the disease than previously reported, though time to recovery might be long in some cases.

**Abstract:**

Acquired canine proximal renal tubulopathy (Fanconi syndrome) related to excessive ingestion of jerky treats has been recognized since 2007. This study aimed to improve knowledge about the syndrome’s characteristics, especially long-term outcome. By reaching out to veterinarians and dog owners, dogs suspected of jerky induced Fanconi syndrome were identified. The dog’s medical records were reviewed, and owners interviewed. Data was analyzed using linear mixed models (*p* < 0.05 was considered statistically significant) and descriptive statistics are reported. Thirty dogs, median body weight 6.8 (range 1.2–59) kg and age 6.5 (0.5–14) years, were enrolled as suspected cases based on history of jerkey ingestion and confirmed normoglycemic/hypoglycemic glycosuria. Clinical signs included polydipsia (23/30), polyuria (21/30), lethargy (19/30), weight loss (15/30), hyporexia (11/30), vomiting (7/30), diarrhea (7/30) and no clinical signs (2/30). Para-clinical findings included azotemia (6/28), hypophosphatemia (9/25), metabolic acidosis (3/8), hypokalemia (6/20), proteinuria (13/26), aminoaciduria (4/4), hematuria (22/29) and ketonuria (7/27). Clinical signs resolved in 22/28 within 11 (0.3–52) weeks and glycosuria resolved in 28/30 within 6.5 (1–31) weeks. There were no associations between serum creatinine and urea and the amount/duration of jerky ingestion. Serum symmetric dimethylarginine concentrations were only available for a few dogs, therefore no conclusion was achieved on a possible association with duration of jerky ingestion. Apart from a larger percentage of dogs achieving complete recovery, the current findings are in agreement with previous reports.

## 1. Introduction

Fanconi syndrome is a proximal renal tubulopathy characterised by renal loss of micronutrients and minerals/electrolytes, most importantly glucose, amino acids, potassium, bicarbonate, and phosphate [1]. These components of the plasma are freely filtered into the ultrafiltrate in the glomerulus and in the healthy individual reabsorbed completely (glucose and amino acids) or partially (minerals and electrolytes) in the proximal tubular system [2,3]. Canine Fanconi syndrome was first described in 1976 [4] and is mostly recognized as a congenital disease in Basenjis [5], though similar conditions have been reported in Norwegian Elkhound [6] and Irish Wolfhounds [7] as well. Single cases of Fanconi syndrome have also been described in a Whippet [8] and a Yorkshire Terrier [9].

Acquired Fanconi syndrome has previously been considered a rare disease, described in cases of copper storage disease [10], lead [11] and ethylene glycol [12] toxicity, primary hypoparathyroidism [13], and associated with gentamycin [14] and amoxicillin [15] treatment. Furthermore, features of the Fanconi syndrome, especially glycosuria, are also well known in canine leptospirosis [16,17] and have been described as comorbidity of acute pancreatitis [18]. However, canine acquired Fanconi syndrome has become much more frequent in the last 15 years with the U.S. Food and Drug Agency (FDA) receiving more than 360 reports of dogs diagnosed with acquired Fanconi syndrome, in the period of 2007–2015 [19]. This increased number of acquired Fanconi syndrome cases is believed to be related to excessive ingestion of jerky treats, most often based on chicken, and originating from China. Although intensive investigation has been carried out, the underlying cause for the jerky induced Fanconi syndrome (JFS) remains unknown [19,20,21].

Clinical signs of JFS are unspecific and include lethargy, decreased appetite, vomiting, polyuria/polydipsia, and weight loss [22,23,24,25,26]. Para-clinical hallmarks for JFS are normoglycemic/hypoglycemic glycosuria. Other common reported biochemical and urine abnormalities are azotemia, metabolic acidosis, hypokalaemia, hypophosphatemia, aminoaciduria, proteinuria, ketonuria, and haematuria (Table 1).

Apart from an Australian retrospective study of 108 dogs [22], current literature mainly consist of case reports/case series. Especially information on outcome and long-term follow-up for dogs diagnosed with JFS is still sparse. 

This retrospective observational study aimed to contribute to the knowledge about syndrome characteristics specifically in relation to recovery and long-term outcome in dogs diagnosed with FJS. 

## 2. Materials and Methods

The study was designed as a retrospective cohort study. Prior to initiation the local administrative and ethical committee at the Department of Veterinary Clinical Sciences, University of Copenhagen, Denmark (approval number 2020-7), ethically approved the study protocol.

Dogs suspected of previous or current JFS were identified by reaching out to Danish veterinarians and owners through social media and national veterinary as well as dog owners’ magazines. 

Responding veterinarians were asked to retrieve a consent from the owner prior to delivery of medical record information, laboratory results as well as owner contact information. Dog owners, who responded directly, were vice versa asked for permission to contact the consulting veterinarian for medical records. 

Medical records and laboratory results were reviewed, and owners answered a standardised questionnaire either in writing or through a telephone interview. The questionnaire detailed diet and treats, time of onset, clinical signs observed, disease development, treatment strategies, time to recovery (if achieved), any ongoing issues, and concurrent medical conditions and therapy. In order to quantify the amount of jerky treats ingested by the dogs, conversions of one teaspoon of small treats to 5 g, one tablespoon of small treats to 15 g, one filet to 30 g and the jerky part of one chew stick to 15 g were applied (Figure 1).

Dogs were included in the study if normoglycemic/hypoglycemic glycosuria had been confirmed. Dogs were excluded if there was no history of ingestion of jerky treats, if the medical records raised a suspicion of an underlying disease and/or medical treatment reported to be associated with Fanconi Syndrome, or if leptospirosis had not been excluded by the attending veterinarian. 

For statistical analysis, the software system R version 4.2.0 (22 April 2022) [31] was used. A separate linear model (lme4 and lmerTest package) was applied with log transformed serum creatinine, urea and symmetric-dimethylarginine (SDMA) concentrations at the time of JFS diagnosis as the outcome, while number of weeks with jerky feeding and estimated daily amount fed were included as fixed effects in each model. Standard diagnostics were performed for each model, for verification regarding distribution assumptions, and detection of observations with undue influence. A Wilcoxon signed rank sum test was used to test if medical treatment (as outpatient or hospitalised) affected time to clinical resolution. Results are stated as median (range) and *p* < 0.05 was considered statistically significant.

## 3. Results

### 3.1. Study Population 

Thirty-five dogs were initially recruited. After reviewing medical records and owner questionnaires, five dogs were excluded due to no history of jerky ingestion (*n* = 2), no medical record confirming glycosuria (*n* = 1), possible leptospirosis (*n* = 1), and suspected familial nephropathy (*n* = 1). Thirty dogs diagnosed with JFS in the period June 2017–April 2020 were finally included. The 30 dogs were reported by 27 veterinarians from 17 different Danish veterinary clinics or hospitals. There were 5 entire females, 4 neutered females, 15 entire males, and 6 neutered males. Twenty-seven of the 30 dogs were small breeds (Cavalier King Charles Spaniel (*n* = 4), Chihuahua (*n* = 4), Jack Russell Terrier (*n* = 3), Cairn Terrier (*n* = 2), Miniature poodle (*n* = 2), West Highland White Terrier (*n* = 2), Shih Tzu (*n* = 2), Havanese (*n* = 2), Coton de Tulear (*n* = 1), Miniature pinscher (*n* = 1), Russian Toy (*n* = 1), Maltese (*n* = 1), Japanese Spitz (*n* = 1), and mixed small breed (*n* = 1)). The remaining 3/30 dogs were large breeds (Great Dane (*n* = 1), Stabyhound (*n* = 1), and large mixed breed (*n* = 1)). The median body weight was 6.8 (1.2–59) kg and median age was 6.5 (0.5–14) years. Pre-existing medical conditions included orthopedic problems (*n* = 3), dermatitis/allergy (*n* = 5), valvular heart disease (*n* = 2), pseudo-pregnancy (*n* = 2), neurological problems (*n* = 2), chronic hepatitis (*n* = 2) chronic gastroenteritis (*n* = 1), cutaneous tumors (*n* = 1), cryptorchidism (*n* = 1) and lower urinary tract infection (*n* = 1). Both dogs diagnosed with hepatitis were diagnosed by histopathology and copper score evaluation excluding copper storage disease. Seventeen of the 30 dogs had received medical treatment (prednisolone (*n* = 6), non-steroidal anti-inflammatory drugs (*n* = 6), opioids (*n* = 4), and sedatives (*n* = 5)) less than 2 months prior to being diagnosed with JFS. 

### 3.2. Clinical Signs

Clinical signs included polydipsia (23/30), polyuria (21/30), lethargy (19/30), weight loss (15/30), hyporexia (11/30), vomiting (7/30), and diarrhea (7/30). Two dogs showed no clinical signs (for further details see Figure 2).

### 3.3. Para-Clinical Signs

All 30 dogs had confirmed normoglycemic/hypoglycemic glycosuria, but the extent of further diagnostic work-up varied. One dog had transient mild hyperglycemia, which later normalized without resolution of glycosuria. Additional para-clinical findings included azotemia (6/28), hypophosphatemia (9/25), metabolic acidosis (3/8), hypokalemia (6/20), proteinuria (13/26), aminoaciduria (4/4), hematuria (22/29), and ketonuria (7/27), (for further details, see Figure 3). 

There was no association between the owner estimated daily jerky allowance and serum creatinine (*p* = 0.8), symmetric dimethylarginine (SDMA) (*p* = 0.8) or urea (*p* = 0.4) concentrations. Nor was there any association between duration of jerky feeding and serum creatinine (*p* = 0.9), and urea (*p* = 0.2) concentrations (Figure 4). For the serum SDMA fewer data were available (*n* = 10). However, while duration of jerky feeding was not significant when all the dogs with SDMA measured at the time of diagnosis were included (*p* = 0.2), a data point from one of the dogs was found to have undue influence on the SDMA regression model. When modeling was performed without this data point an association between serum SDMA concentration at the time of JFS diagnosis and duration of jerky feeding in weeks was statistically significant (*p* = 0.04).

### 3.4. Treatment and Outcome 

For 12/30 dogs, discontinuation of jerky feeding was the only treatment, while 11/30 received supportive medical treatment as outpatients: antiemetics, angiotensin-converting enzyme inhibitors, antibiotics (until leptospirosis could be excluded), renal diet, oral supplementation of phosphate, bicarbonate, folic acid, and cobalamin. In addition, 7/30 dogs were hospitalised for a median of 4 (1–9) days, primarily for intravenous fluid treatment and tube feeding. Medical treatment (as outpatient or hospitalised) did not affect time to clinical resolution (*p* = 0.8). 

Clinical signs resolved completely in 22 of the 28 dogs reported to have clinical signs (Figure 5). Time to recovery from clinical signs was 11 (0.3–52) weeks. Five out of 28 dogs were reported with ongoing clinical signs (see Table 2 for details), while one dog was lost to follow-up. This dog initially showed improvement in clinical signs but was reported deceased for unknown reasons 2–3 months later. Glycosuria resolved in 28/30 dogs within 6.5 (1–31) weeks (Figure 5). On dog had persistent glycosuria and for one dog resolution of glycosuria was unknown as no follow-up on urine analysis was available (Table 2). 

## 4. Discussion

To the authors’ knowledge, this study is one of the largest studies concerning clinical presentation and outcome in dogs diagnosed with JFS. In addition, this study examines recovery and long-term outcome to further qualify the clinical prognosis. One other retrospective study reported a larger number of cases of JFS (*n* = 108), but for most of these dogs, long-term follow-up was either not available or was stated to be ongoing. Time to resolution of clinical signs and glycosuria was reported for only 35 and 8 dogs, respectively [22]. 

In this retrospective observational study, 30 Danish dogs diagnosed with JFS in the period of June 2017–April 2020 were identified. In recent years, JFS has been given less attention, with FDA reporting a marked decrease in reported cases since 2015 [19,20]. Apart from a new case series from Germany published in 2021 [30], all case reports known to the authors have been published from 2011–2017 (*n* = 8). The present study, together with the recent German study, indicate that despite less cases are being reported to US authority and published in the scientific literature, JFS still occur in the canine population and there is a need for continuous awareness of the syndrome. 

In the present study, the clinical signs most frequently mentioned were polyuria, polydipsia, lethargy, weight loss, hyporexia, vomitus, and diarrhea, which is similar to previous reports [22,23,24,25,26,27,28,29,30]. Para-clinical findings included aminoaciduria, hematuria, proteinuria, metabolic acidosis, hypophosphatemia, hypokalemia, ketonuria, and azotemia, and were comparable to earlier studies, both in nature and frequency of appearance (Table 1). 

Interestingly, the majority of dogs in the present study recovered fully in terms of clinical signs (79%) and glycosuria (93%). This outcome is more favorable than expected based on the previous retrospective study, where resolution of clinical signs was reported for 34% (35/102 dogs) and resolution of glycosuria was reported in 12 out of 28 dogs (43%) for which long term follow-up was available [22]. The higher recovery rate reported in the current study might reflect longer follow-up periods (up to 72 weeks) for many dogs, though comparison with the above mentioned study is difficult as follow-up periods were not stated directly in the publication [22]. The time span for clinical recovery was from less than a week up to a year, while glycosuria resolved within 1–31 weeks. Therefore, the longer follow-up period for most dogs in the current study is most likely the reason for identifying a larger proportion of fully recovered dogs, compared to previous studies. It is possible therefore that this current study better reflects the true outcome. However, the true recovery rate might be even higher, as some of the dogs, categorized as ongoing had their last follow-up at 8 or 9 weeks after diagnosis, which is less than median time to recovery.

Interestingly, the recovery from glycosuria was markedly faster than complete recovery of clinical signs. For most of the previously published cases, it is not clear whether clinical signs or glycosuria resolved first, though in one case report glycosuria was reported to resolve ahead of clinical signs [25]. This is in contrast to two case series of 2 dogs [26] and 1 dog [29], respectively, in which all other clinical signs resolved first. 

There was no association between the owner estimated daily jerky allowance and any of the glomerular filtration biomarkers, nor between the serum creatinine or urea and the duration of jerky feeding. The results were less clear for SDMA as only 10 dogs had serum SDMA concentrations measured at time of JFS diagnosis. One data point was found to overtly influence the model more than any other data point. When the model was performed without this data point a significant association between serum SDMA at the time of JFS diagnosis and duration of jerky feeding. This is a sign of lack of statistical power as one data point should not change the significance of a result, and as a consequence a conclusion on any potential association between duration of jerky feeding and serum SDMA concentrations could not be reached for this study. 

Azotemia was only present in 6 of the 28 dogs, for which this information was available. Furthermore, there was no association between azotemia and the daily allowance of jerky treats or the duration of jerkey feeding. The lacking association related to creatinine and urea concentrations indicate that the renal disease induced in dogs by excessive ingestion of jerky treats does not significantly affect the glomerulus, which supports the previously reported necropsy results, in which, renal damage was primarily located to the proximal tubules [22]. However, as the possibility of an association between duration of feeding jerky and SDMA still remains and considering that SDMA is generally considered a more sensitive marker of glomerular damage than creatinine [32,33,34], SDMA should be further evaluated in future prospective JFS studies. 

For two dogs, CKD, international renal interest society stage 1 or 2, and decreased glomerular filtration rate was diagnosed following the JFS diagnosis. For both dogs, renal status prior to the debut of JFS was unknown and it was unclear whether CKD was pre-existing or a consequence of JFS. While not all dogs, who ingest jerky treats develop JFS, it could be interesting for future studies to examine if pre-existing CKD can be a predisposing factor for developing JFS. 

The retrospective nature of the current study gave the opportunity for a long follow-up time on clinical signs. This was important in relation to investigating outcome, as the clinical signs of JFS seem to persist for a long time in some individuals, as reported in a previous case [22]. However, a limitation of the current study is, that the time to recovery, in some cases, was not clearly stated in the medical record. In such cases the reported time to recovery of clinical signs depended on owner recollection. Furthermore, because some of the JFS cases had an onset several years prior to the time of owner interview, some information might not be accurately recalled. In particular, quantification of jerky treats proved challenging in such cases All owners fed jerkey to their dogs in good faith, with no suspicion of the treat being problematic and therefore did they not pay close attention to the amount fed. Adding this to the time passed since they fed the treat, the estimates of quantification have to be interpreted as approximate amounts. With the commonly long course of the JFS in mind, future studies of the disease should preferably be planned as prospective long-term studies in order to preserve details as accurate as possible. 

## 5. Conclusions

In conclusion, this study found that JFS has a good long-term prognosis for most dogs, but that the time to full clinical recovery might be longer than previously reported. Most dogs exhibit nephropathy located to the proximal tubules without affecting glomerular filtration, as supported by lack of association between serum creatinine and urea concentrations and duration and amount of jerky ingestion. For future studies, a prospective study design with enrolment of affected dogs at time of diagnosis, would be preferable to better preserve details and ensure uniform data collection.

## Figures and Tables

**Figure 1 animals-12-03192-f001:**
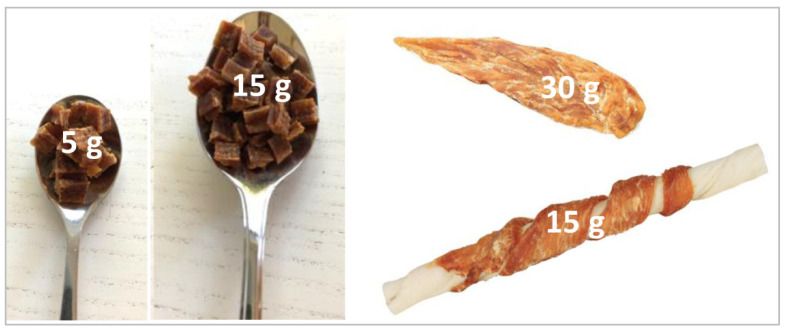
Quantification of jerky treats. Conversions applied for the amount of jerky ingested by the dogs as stated by the owner in a questionnaire for owners of dogs suspected of jerky induced Fanconi syndrome.

**Figure 2 animals-12-03192-f002:**
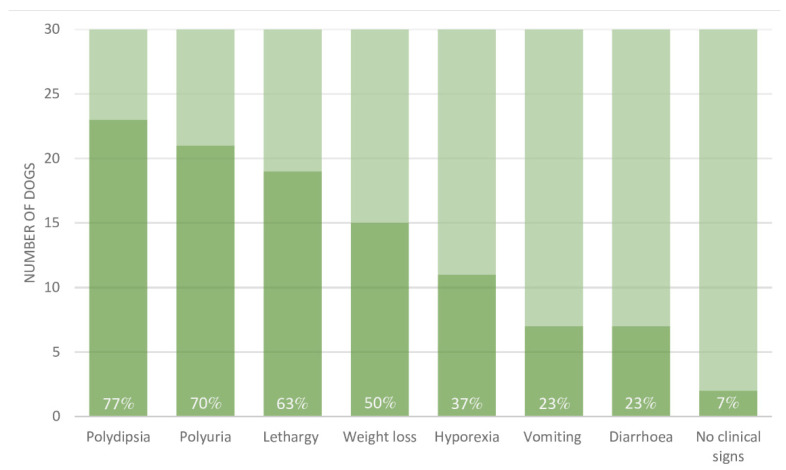
Clinical signs in 30 dogs diagnosed with jerky induced Fanconi syndrome. The dark portion of each bar represent the proportion of dogs reported to display the clinical sign in question out of the 30 dogs included (the total height of each bar). The prevalence for each clinical sign is indicated in percentage as well.

**Figure 3 animals-12-03192-f003:**
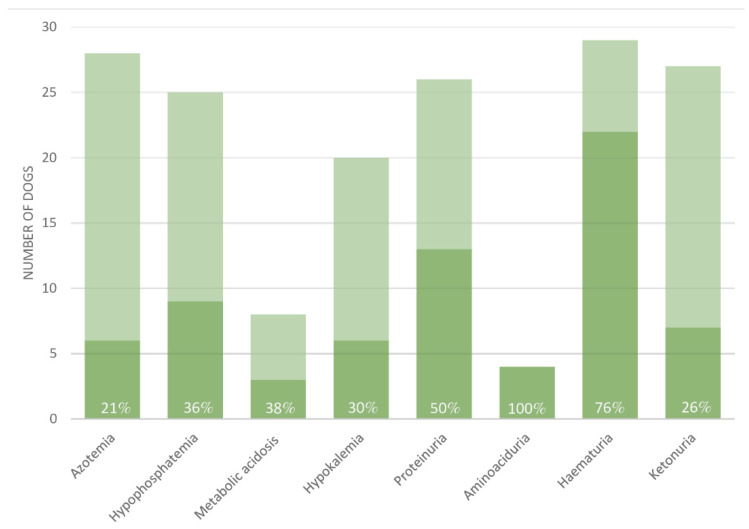
Para-clinical findings in 30 dogs diagnosed with jerky induced Fanconi syndrome. Para-clinical findings reported in medical records from attending veterinarians for 30 dogs diagnosed with acquired Fanconi syndrome associated with ingestion of jerky treats. All dogs were included due to confirmed normoglycemic/hypoglycemic glycosuria. The dark color represents the number of dogs reported positive for the finding in question. The total height of each bar represents the total number of dogs for which the specific para-clinical finding had been investigated. The white numbers provide the percentage of positive findings.

**Figure 4 animals-12-03192-f004:**
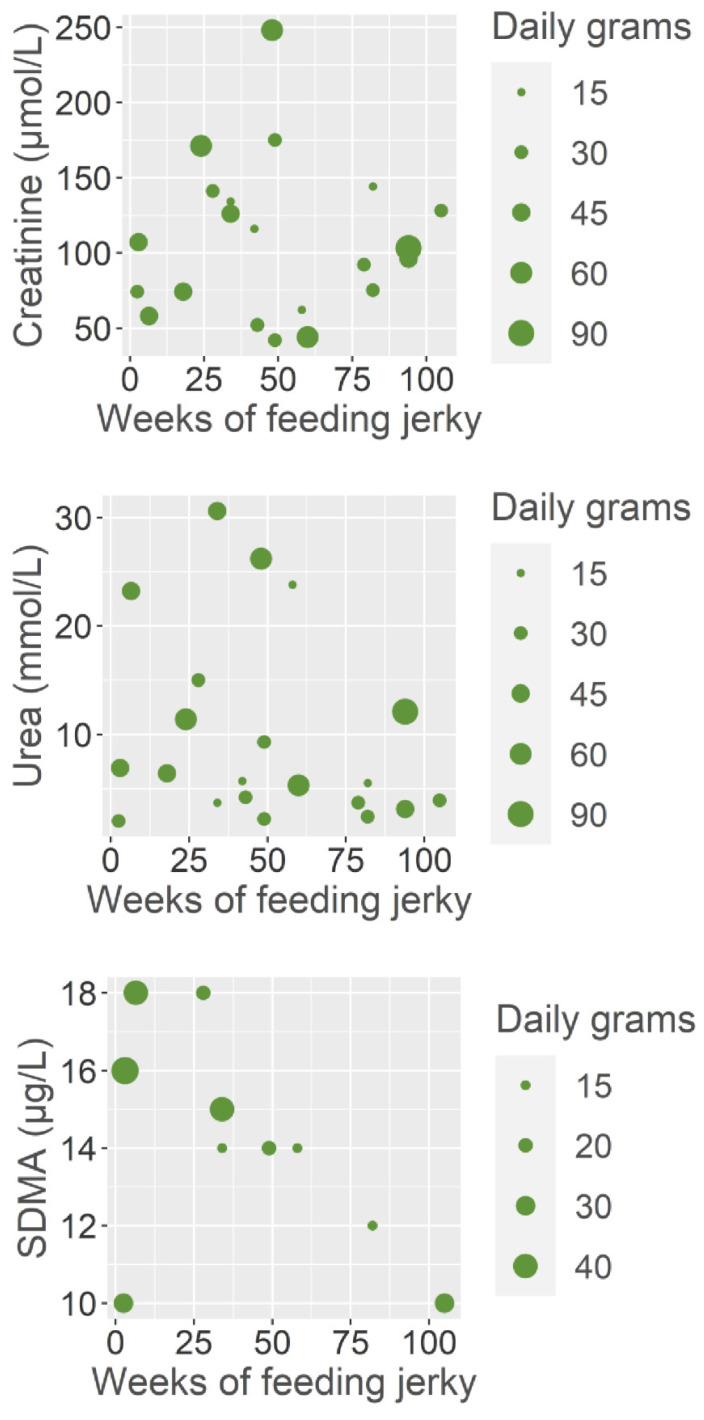
Bubble chart between markers of glomerular function, estimated duration and amount of jerky ingestion in dogs diagnosed with jerky induced Fanconi syndrome. On the *y*-axis the concentration of the glomerular function marker (serum creatinine (µmol/L), symmetric dimethylarginine (SDMA) (µg/L), and urea (mmol/L)), and on the *x*-axis the estimated duration of jerky ingestion (weeks). The size of the dots differ depending on the estimated amount of daily jerky fed to each dog. Creatinine and urea data was available from 21 dogs, while only 10 dogs had SDMA concentration measured at the time of diagnosis.

**Figure 5 animals-12-03192-f005:**
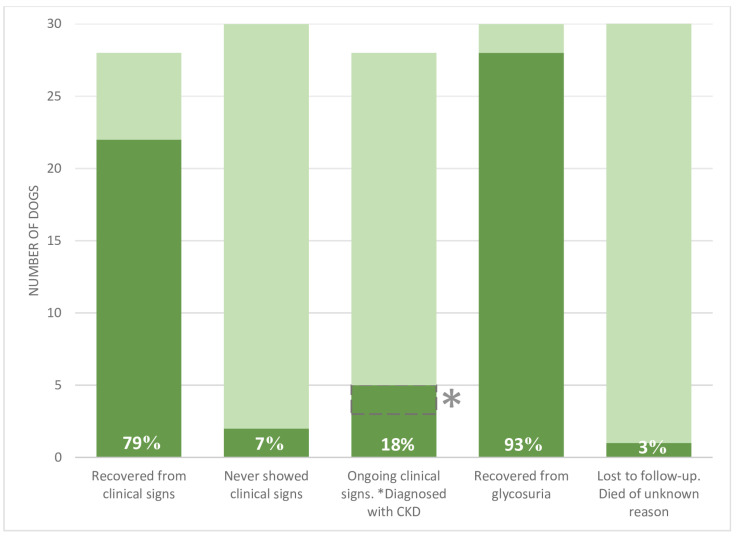
Clinical and para-clinical outcome of 30 dogs diagnosed with jerky induced Fanconi syndrome. Outcome of 30 dogs diagnosed with acquired Fanconi syndrome associated with ingestion of jerky treats. Dark color represents the number of dogs reported positive for the finding in question. The total height of each bar represents the total number of dogs for which the specific statement is relevant. The white numbers provide the percentage of positive findings. * Out of the 5 dogs reported to have continued clinical signs, 2 were diagnosed with chronic kidney disease (CKD) based on International Renal Interest Society guidelines.

**Table 1 animals-12-03192-t001:** Previously reported para-clinical abnormalities and outcome of jerky induced Fanconi syndrome.

Origin	Number of Dogs	Azotaemia	Metabolic Acidosis	Hypokalaemia	Hypo-Phosphatemia	Glycosuria	Aminoaciduria	Proteinuria	Ketonuria	Haematuria
USA, 2011 [23]	4	25%	100%	100%	25%	100%	100%	100%	50%	75%
Australia, 2013 [22]	108	27%	77%	45%	37%	100%	100%	87%	25%	77%
Switzerland, 2014 [24]	1		+	+	+	+	NA	+	+	-
UK, 2016 [27]	11	-	NA	NA	NA	+	+	-	NA	NA
UK, 2014 [28]	1	-	-	-	-	+	+	-	-	-
Austria, 2015 [29]	1	-	NA	NA	+	+	+	+	-	+
Japan, 2015 [25]	1	+	+	+	NA	+	+	+	NA	-
Japan, 2017 [26]	2	-	50%	50%	50%	100%	100%	50%	50%	-
Germany, 2021 [30]	6	83%	NA	33%	0%	100%	100%	67%	NA	50%

+: reported present, -: reported absent, NA: not assessed, PU: polyuria, PD: polydipsia.

**Table 2 animals-12-03192-t002:** Dogs still showing clinical signs or continued glycosuria at time of study termination.

Dog ID	Clinical Signs	Glycosuria	Subsequent CKD * (+/−)	Last Follow-Up
1	Ongoing 68 wks	Resolved 9 wks	+	68 wks
4	Ongoing 35 wks	Resolved 22 wks	+	35 wks
18	Ongoing 8 wks	Unknown	−	8 wks
23	Ongoing 72 wks	Resolved (unknown time)	−	72 wks
32	Ongoing 9 wks	Resolved 4 wks	−	9 wks
2	Recovered 20 wks	Ongoing 20 wks	−	20 wks

Out of 30 dogs diagnosed with jerky induced Fanconi syndrome, 6 dogs had ongoing clinical signs and/or continued glycosuria at time of study termination. * Two of these dogs had persistent azotaemia after resolution of glycosuria and were hereafter diagnosed with chronic kidney disease (CKD) stage 1 and 2, respectively, based on International Renal Interest Society guidelines. wks: weeks since diagnosed with jerky induced Fanconi syndrome.

## Data Availability

The datasets used and analyzed during the current study are available at UCPH ERDA https://sid.erda.dk/sharelink/dxz1jTAHlm (accessed on 19 October 2022).

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
