# Peer review of "Outcome of Acquired Fanconi Syndrome Associated with Ingestion of Jerky Treats in 30 Dogs"

_animals, 2022, doi:10.3390/ani12223192_

Round 1

Reviewer 1 Report

The manuscript is well written and presented, the results are interesting especially for the veterinarians. This paper seems like a case report.  

There are a lack of informations concerning the reporting of veterinarians, for exemple,  how many veterinarian were interviewed.

The authors have to add some informations, like; the period of the appearance of symptoms since the first ingestion of jerky treats. 

The authors should add some informations about dogs before contracting JKS (anamnesis).

The authors should add perspectives in the conclusion 

Reviewer 2 Report

This paper which is called "Outcome of acquired Fanconi syndrome associated with ingestion of jerky treats in 30 dogs" is a very good scientific paper and well written. Therefore I accepted this version of the paper.

1. This is a retrospective study and this review contains Fanconi Syndrome. Normally this disease is a rare one. Therefore 30 of them are very good cases. This study aimed to improve knowledge 25 about disease characteristics, especially long-term outcomes.

2. I consider the topic original, because Fanconi syndrome is a very important disease in dogs and some practitioners can not diagnose of this disease.

3. Normally, we can not find the number of Fanconi cases. Writers use many of the details of these dogs.

4.  Writers can use many of clinical signs resolved in 22/28 within 11 (0.3-52) 35 weeks and glycosuria resolved in 28/30 within 6.5 (1-31) weeks.

5. At the end of the study they find good relations with the aim of the review.

6. References are appropriate.

7. Researchers did a good search about literature and organized understandable tables and figures.

Reviewer 3 Report

The title, abstract, and keywords:  adequately represent the study.

The introduction briefly presents the disease entity. 

The following sections describe clearly and scrupulously the possibilities of assessing Fanconi syndrome.

The tables and figures well represent the data about Fanconi syndrome. 

In lines 49 and 51 there is a lack information about Whippet and Yorkshire terrier. 

(1. MacKenzie CP, van den Broek A. The Fanconi syndrome in a Whippet. J Small Anim Pract 1982;23(8):469–74.

2. McEwan NA, Macartney L. Fanconi’s syndrome in a Yorkshire terrier. J Small Anim Pract 1987;28(8):737–42).

In section: treatment - lack information about angiotensin-converting enzyme (ACE). 

Reviewer 4 Report

The manuscript is well structured and can represent an excellent addition to the existing bibliography on Fanconi syndrome in dogs related to ingestion of treats.

Line 106: Authors should explain in the inclusion criteria how the condition of normoglycemia or hypoglycemia was confirmed (e.g. how many samples in how long, fructosamines). Diseases associated with the transient acquired form of canine Fanconi syndrome include hypoparathyroidism, proximal renal tubular acidosis, drug-or toxin-induced injury, and systemic hypertension, for this reason the authors should better explain whether these pathologies were excluded in the population with specific test. A study showed transient acquired fanconi syndrome  with copper storage hepatopathy. Copper storage hepatopathies have been diagnosed in breeds such as West Highland White Terrier. The authors should explain if this pathology was excluded in the two purebred dogs (West Highland White Terrier).

Line 109: The authors should explain with which method the positivity for Leptospirosis infection was excluded.
